# Reinforcement Learning with Combinatorial Actions: An Application to Vehicle Routing

**Arthur Delarue**
MIT Operations Research Center
Cambridge, MA
adelarue@mit.edu

**Ross Anderson**
Google Research
Cambridge, MA
rander@google.com

**Christian Tjandraatmadja**
Google Research
Cambridge, MA
ctjandra@google.com

## Abstract

Value-function-based methods have long played an important role in reinforcement learning. However, finding the best next action given a value function of arbitrary complexity is nontrivial when the action space is too large for enumeration. We develop a framework for value-function-based deep reinforcement learning with a combinatorial action space, in which the action selection problem is explicitly formulated as a mixed-integer optimization problem. As a motivating example, we present an application of this framework to the capacitated vehicle routing problem (CVRP), a combinatorial optimization problem in which a set of locations must be covered by a single vehicle with limited capacity. On each instance, we model an action as the construction of a single route, and consider a deterministic policy which is improved through a simple policy iteration algorithm. Our approach is competitive with other reinforcement learning methods and achieves an average gap of 1.7% with state-of-the-art OR methods on standard library instances of medium size.

## 1 Introduction

Reinforcement learning (RL) is a powerful tool that has made significant progress on hard problems. For instance, in games like Atari [1] or Go [2], RL algorithms have devised strategies that significantly surpass the performance of experts, even with little to no knowledge of problem structure. The success of reinforcement learning has carried over to other applications such as robotics [3] and recommender systems [4]. As they encompass ever more application areas, RL algorithms need to be adjusted and expanded to account for new problem-specific features.

One area in which RL has yet to make a convincing breakthrough is combinatorial optimization. There has been significant effort in recent years to apply RL frameworks to NP-hard combinatorial optimization problems [5, 6, 7], including the traveling salesman problem (TSP) or more general vehicle routing problem (VRP) (see [8] for a recent survey). Solving these problems produces a practical impact for many industries, including transportation, finance, and energy. Yet in contrast to [1, 2], RL methods have yet to match the performance of expert implementations in this domain, such as the state-of-the-art Concorde TSP solver [9, 10].

A major difficulty of designing an RL framework for combinatorial optimization is formulating the action space. Value-based RL algorithms typically require an action space small enough to enumerate, while policy-based algorithms are designed with continuous action spaces in mind [11]. These twin requirements severely limit the expressiveness of the action space, thus placing the entire weight of the problem on the machine learning model. For instance, in [12], the selected action space models advancing the current vehicle one city at a time. RL algorithms for combinatorial optimization must therefore rely on complex architectures such as pointer networks [5, 6, 12] or graph embeddings [7].

This paper presents a different approach, in which the combinatorial complexity is captured not only by the learning model, but also by the formulation of the underlying decision problem. We focus on the Capacitated Vehicle Routing Problem, a classical combinatorial problem from operations research [13], where a single capacity-limited vehicle must be assigned one or more routes to satisfy customer demands while minimizing total travel distance. Despite its close relation to the TSP, the CVRP is a much more challenging problem and optimal approaches do not scale past hundreds of cities. Our approach is to formulate it as a sequential decision problem where the state space is the set of unvisited cities, and the action space consists of feasible routes. We estimate the value function of a state (i.e., its cost-to-go) using a small neural network. The action selection problem is then itself combinatorial, with the structure of a Prize Collecting Traveling Salesman Problem (PC-TSP) with a knapsack constraint and a nonlinear cost on unvisited cities. Crucially, the PC-TSP is a much easier combinatorial problem than the CVRP, allowing us to tractably exploit some of the combinatorial structure of the problem.

The contributions in this paper are threefold.

1. We present a policy iteration algorithm for value-based reinforcement learning with combinatorial actions. At the *policy improvement* step, we train a small neural network with ReLU activations to estimate the value function from each state. At the *policy evaluation* step, we formulate the action selection problem from each state as a mixed-integer program, in which we combine the combinatorial structure of the action space with the neural architecture of the value function by adapting the branch-and-cut approach described in [14].

2. We apply this technique to develop a reinforcement learning framework for combinatorial optimization problems in general and the Capacitated Vehicle Routing Problem in particular. Our approach significantly differs from existing reinforcement learning algorithms for vehicle routing problems, and allows us to obtain comparable results with much simpler neural architectures.

3. We evaluate our approach against several baselines on random and standard library instances, achieving an average gap against the OR-Tools routing solver of 1.7% on moderately-sized problems. While we do not yet match state-of-the-art operations research methods used to solve the CVRP, we compare favorably with heuristics using oracles of equal strength to our action selection oracle and to other RL approaches for solving CVRP [12].

## 2   Background and related work

**Combinatorial action spaces.** Discrete, high-dimensional action spaces are common in applications such as natural language processing [15] and text-based games [16], but they pose a challenge for standard RL algorithms [17], chiefly because enumerating the action space when choosing the next action from a state becomes impossible. Recent remedies for this problem include selecting the best action from a random sample [15], approximating the discrete action space with a continuous one [18, 19], or training an additional machine learning model to wean out suboptimal actions [16]. None of these approaches guarantees optimality of the selected action, and projection-based approaches in particular may not be applicable when the structure of the action space is more complex. In fact, even continuous action spaces can prove difficult to handle: for example, safety constraints can lead to gradient-based methods providing infeasible actions. Existing ways to deal with this issue include enhancing the neural network with a safety layer [20], or modifying the policy improvement algorithm itself [21]. More recently, [22] propose a framework to explicitly formulate the action selection problem using optimization in continuous settings. We note that though combinatorial action spaces pose a challenge in deep reinforcement learning, they have been considered before in approximate dynamic programming settings with linear or convex learners [23]. In this paper, we formulate and solve the problem of selecting an optimal action from a combinatorial action space using mixed-integer optimization.

**Combinatorial optimization and reinforcement learning.** In recent years, significant work has been invested in solving NP-hard combinatorial optimization problems using machine learning, notably by developing new architectures such as pointer networks [5] and graph convolutional networks [24]. Leveraging these architectures, reinforcement learning approaches have been developed for the TSP [6, 7] and some of its vehicle routing relatives [25], including the CVRP [12]. Crucially, the CVRP is significantly more challenging than the closely related TSP. While TSPs on tens of thousands

of cities can be solved to optimality [26], CVRPs with more than a few hundred cities are very hard to solve exactly [27], often requiring cumbersome methods such as branch-and-cut-and-price, and motivating the search for alternative solution approaches. This work adopts a hybrid approach, casting a hard combinatorial problem (CVRP) as a sequence of easier combinatorial problems (PC-TSP) in an approximate dynamic programming setting.

**Optimizing over trained neural networks.** A major obstacle for solving the action selection problem with an exponential discrete action space is the difficulty of finding a global extremum for a nonlinear function (as a neural network typically is) over a nonconvex set. One possible solution is to restrict the class of neural networks used to guarantee convexity [28], but this approach also reduces the expressiveness of the value function. A more general approach is to formulate the problem as a mixed-integer program (MIP), which can be solved using general-purpose algorithms [14]. Using an explicit optimization framework allows greater modeling flexibility and has been successfully applied in planning settings [29, 30] as well as in RL problems with a continuous action space [22]. Mixed-integer optimization has also proven useful in neural network verification [31, 32]. In this paper, we use neural networks with ReLU activations, leveraging techniques developed in [14] to obtain a strong formulation of the action selection problem.

# 3  Reinforcement learning model for CVRP

## 3.1  Problem formulation

A CVRP instance is defined as a set of $n$ cities, indexed from 0 to $n-1$. Each city $i > 0$ is associated with a demand $d_i$; the distance from city $i$ to city $j$ is denoted by $\Delta_{ij}$. City 0 is called the depot and houses a single vehicle with capacity $Q$ (or equivalently, a fleet of identical vehicles). Our goal is to produce routes that start and end at the depot such that each non-depot city is visited exactly once, the total demand $d_i$ in the cities along one route does not exceed the vehicle capacity $Q$, and the total distance of all routes is minimized. We assume the number of routes we can serve is unbounded, and note that the distance-minimizing solution does not necessarily minimize the number of vehicles used.

We formulate CVRP as a sequential decision problem, where a *state $s$* corresponds to a set of as-yet-unvisited cities, and an *action $a$* is a feasible route starting and ending at the depot and covering at least one city. We represent states using a binary encoding where 0 indicates a city has already been visited and 1 indicates it has not, i.e., $\mathcal{S} = \{0, 1\}^n$ (though by convention, we never mark the depot as visited). The action space $\mathcal{A}$ corresponds to the set of all partial permutations of $n-1$ cities. Note that the sizes of both the state space and the action space are exponential in $n$, even if we only consider actions with the shortest route with respect to their cities.

The dynamics of the decision problem are modeled by a deterministic transition function $T : \mathcal{S} \times \mathcal{A} \to \mathcal{S}$, where $T(s, a)$ is the set of remaining unvisited cities in $s$ after serving route $a$, and a cost function $C : \mathcal{A} \to \mathbb{R}$, indicating the cost incurred by taking action $a$. Since cities cannot be visited twice, $T(s, a)$ is undefined if $a$ visits a city already marked visited in $s$. For clarity we define $\mathcal{A}(s) \subseteq \mathcal{A}$ as the set of feasible actions from state $s$. The unique terminal state $s^{\text{term}}$ corresponds to no remaining cities except the depot. Finding the best CVRP solution is equivalent to finding a least-cost path from the initial state $s^{\text{start}}$ (where all cities are unvisited) to $s^{\text{term}}$.

We consider a deterministic policy $\pi : \mathcal{S} \to \mathcal{A}$ specifying the action to be taken from state $s$, and we let $\Pi$ designate the set of all such policies. The value of a state $V^\pi(s)$ is the cost incurred by applying $\pi$ repeatedly starting from state $s$, i.e., $V^\pi(s) = \sum_{t=1}^{T} C(a^t | a^t = \pi(s^{t-1}), s^t = T(s^{t-1}, a^t), s^0 = s, s^T = s^{\text{term}})$. An optimal policy $\pi^*$ satisfies $\pi^* = \arg\min_{\pi \in \Pi} V^\pi(s^{\text{start}})$.

Given a starter policy $\pi_0$, we repeatedly improve it using a simple *policy iteration* scheme. In the $k$-th *policy evaluation* step, we repeatedly apply the current policy $\pi_{k-1}(\cdot)$ from $N$ randomly selected start states $\{s^{0,i}\}_{i=1}^{N}$. For each random start state $s_{0,i}$, we obtain a *sample path*, i.e. a finite sequence of action-state pairs $(a^{1,i}, s^{1,i}), \ldots, (a^{T,i}, s^{T,i})$ such that $s^{t,i} = T(s^{t-1,i}, a^{t,i})$, and $s^{T,i} = s^{\text{term}}$, as well as the cumulative cost $c^{t,i} = \sum_{t'=t}^{T} C(a^{t',i})$ incurred from each state in the sample path. In the *policy improvement* step, we use this data to train a small neural network $\hat{V}^{k-1}(\cdot)$ to approximate the

value function $V^{\pi_{k-1}}(\cdot)$. This yields a new policy $\pi_k(\cdot)$ defined using the Bellman rule:

$$\pi_k(s) = \arg \min_{a \in \mathcal{A}(s)} C(a) + \hat{V}^{k-1}(t := T(s,a)). \tag{1}$$

## 3.2 Value function learning

The approximate policy iteration approach described above is both on-policy and model-based:[1] in each iteration, our goal is to find a good approximation for the value function $V^\pi(\cdot)$ of the current policy $\pi(\cdot)$. The value function $V^\pi(s)$ of a state $s$ is the cost of visiting all remaining cities in $s$, i.e., the cost of solving a smaller CVRP instance over the unvisited cities in $s$ using policy $\pi(\cdot)$.

In our approximate dynamic programming approach, the value function captures much of the combinatorial difficulty of the vehicle routing problem, so we model $\hat{V}$ as a small neural network with a fully-connected hidden layer and rectified linear unit (ReLU) activations. We train this neural network to minimize the mean-squared error (MSE) on the cumulative cost data gathered at the policy evaluation step. To limit overfitting, we randomly remove some data (between 10% and 20%) from the training set and use it to evaluate the out-of-sample MSE.

We select initial states by taking a single random action from $s^{\text{start}}$, obtained by selecting the next city uniformly at random (without replacement) until we select a city that we cannot add without exceeding vehicle capacity. If this procedure does not allow for enough exploration, we may gather data on too few states and overfit the value function. We therefore adapt a technique from approximate policy iteration [33, 34], in which we retain data from one iteration to the next, and exponentially decay the importance of data gathered in previous iterations [35]. Calling $(s^{(i,k')}, c^{(i,k')})$ the $i$-th data point (out of $N_{k'}$) from iteration $k'$, the training objective in iteration $k$ becomes $\sum_{k'=0}^{k} \sum_{i=1}^{N_{k'}} \gamma^{k-k'} (\hat{V}(s^{(i,k')}) - c^{(i,k')})^2$, where $\gamma \in (0,1]$ denotes the *retention factor*.

## 3.3 Policy evaluation with mixed-integer optimization

The key innovation in our approach comes at the policy evaluation step, where we repeatedly apply policy $\pi_k(\cdot)$ to random start states until we reach the terminal state. Applying policy $\pi_k(\cdot)$ to state $s$ involves solving the optimization problem in Eq. (1), corresponding to finding a capacity-feasible route minimizing the sum of the immediate cost (length of route) and the cost-to-go (value function of new state). This is a combinatorial optimization problem (PC-TSP) that cannot be solved through enumeration but that we can model as a mixed-integer program. Without loss of generality we consider problem (1) for a state $s$ such that $s_i = 1$ for $1 \leq i \leq m$, and $s_i = 0$ for $m < i < n$, i.e. such that only the first $m$ cities remain unvisited. We then obtain the following equivalent formulation for problem (1):

$$\min \quad \sum_{i=0}^{m} \sum_{j=0;j \neq i}^{m} \Delta_{ij} x_{ij} + \hat{V}(t) \tag{2a}$$

$$\text{s.t.} \quad \sum_{j=0;j \neq i}^{m} x_{ij} = y_i \qquad \forall\, 0 \leq i \leq m \tag{2b}$$

$$\sum_{j=0;j \neq i}^{m} x_{ji} = y_i \qquad \forall\, 0 \leq i \leq m \tag{2c}$$

$$\sum_{i=1}^{m} d_i y_i \leq Q \tag{2d}$$

$$\sum_{i \in S} \sum_{j \in \{0,\ldots,m\} \setminus S} x_{ij} \geq y_i \qquad \forall\, S \subseteq \{1,\ldots,m\}, i \in S \tag{2e}$$

$$\sum_{i \in S} \sum_{j \in \{0,\ldots,m\} \setminus S} x_{ji} \geq y_i \qquad \forall\, S \subseteq \{1,\ldots,m\}, i \in S \tag{2f}$$

$$\tag{2g}$$

$$t_i = \begin{cases} 1 - y_i, & 0 \le i \le m \\ 0, & m < i < n \end{cases} \tag{2h}$$

$$x_{ij} \in \{0, 1\} \qquad \forall\, 0 \le i \neq j \le m \tag{2i}$$

$$y_0 = 1 \tag{2j}$$

$$y_i \in \{0, 1\} \qquad \forall\, 1 \le i \le m. \tag{2k}$$

The binary decision variable $y_i$ is 1 if city $i$ is included in the route, and 0 otherwise, and the binary decision variable $x_{ij}$ is 1 if city $i$ immediately precedes city $j$ in the route (we constrain $y_0$ to be 1 because the depot must be included in any route). The binary decision variable $t_i$ is 1 if city $i$ remains in the new state $T(s, a)$ and 0 otherwise. The objective (2a) contains the same two terms as (1), namely the total distance of the next route, plus the future cost associated with leaving some cities unvisited.

Constraint (2b) (resp. (2c)) ensures that if city $i$ is included in the route, it must be immediately followed (resp. preceded) by exactly one other city (flow conservation constraints). Constraint (2d) imposes that the total demand among selected cities does not exceed the vehicle's capacity. Constraints (2e) and (2f) ensure that the route consists of a single tour starting and ending at the depot; they are called *cycle-breaking* or *cutset* constraints and are a standard feature of MIP formulations of routing problems. Finally, constraints (2h) relate the action selection variable $y_i$ for each city $i$ to the corresponding new state variables $t_i$.

We make two further comments regarding problem (2). First, we note that the formulation includes an exponential number of constraints of type (2e) and (2f). This issue is typically addressed via "lazy constraints", a standard feature of mixed-integer programming solvers such as Gurobi or SCIP, in which constraints are generated as needed. In practice, the problem can be solved to optimality without adding many such constraints. Second, as currently formulated, problem (2) is not directly a mixed-integer linear program, because $\hat{V}(t)$ is a nonlinear function. However, because we choose ReLU-activated hidden layers, it turns out it is a piecewise linear function which can be represented using additional integer and continuous variables and linear constraints [29, 14, 22].

In the worst case, problem (2) can be solved in exponential time in $m$. But modern MIP solvers such as Gurobi and SCIP use techniques including branch-and-bound, primal and dual heuristics, and cutting planes to produce high-quality solutions in reasonable time [36, 37]. Given enough time, MIP solvers will not just return a solution but also certify optimality of the selected action.

A MIP approach for action selection was presented in [22], but ours differs in several key ways. First, we consider a discrete action space instead of a continuous one. Second, we estimate the cost-to-go directly from cumulative costs rather than using a Q-learning approach because (i) state transitions in our problem are linear in our action and (ii) computing cumulative costs allows us to avoid solving the combinatorial action selection problem at training time, reducing overall computation. Separating training and evaluation may yield additional computational benefits, since the training loop could benefit from a gradient-descent-friendly architecture and hardware such as a GPU/TPU, while the evaluation loop relies on CPU-bound optimization solvers. Separating the tasks allows for better parallelization and hardware specification.

To further take advantage of the combinatorial structure of the problem, we augment the objective (2a) with known lower bounds. The cost-to-go of a state cannot be lower than the distance required to serve the farthest remaining city from the depot, nor can it be exceeded by the summed lengths of each remaining city's shortest incoming edge. Given such lower bounds $\{LB^p(t)\}_{p=1}^P$, we replace $\hat{V}(t)$ in the objective with $\max(\hat{V}, LB^1(t), \ldots, LB^P(t))$. As long as the lower bounds are linear in $t$ (as are the ones we mention), the addition does not significantly increase solve time for problem (2).

## 4 Computational results

We now present results on benchmark instances from the operations research literature and on random instances from [12]. We compare our results to several other approaches, and analyze their sensitivity to input parameters. Our implementation is in C++ and we use SCIP 6.0.2 as a MIP solver.

Table 1: Comparison of results with existing approaches: a "greedy" approach in which each action is selected to minimize immediate distance traveled plus a trivial upper bound on the cost-to-go; results from Nazari et al. [12] and Kool et al. [25]; our own approach (RLCA) with 16 neurons; OR-Tools routing solver with 60s of guided local search (300s for $n = 51$); and an optimal approach using the OR-Tools CP-SAT constraint programming solver (does not solve to optimality within 6 hours for $n = 51$). We report the mean total distance ($\mu$) and standard error on the mean $\sigma_\mu$, over 1000 random instances for each value of $n$.

| Method | $n = 11$ | | $n = 21$ | | $n = 51$ | |
|---|---|---|---|---|---|---|
| | $\mu$ | $\sigma_\mu$ | $\mu$ | $\sigma_\mu$ | $\mu$ | $\sigma_\mu$ |
| Greedy | 4.90 | 0.03 | 7.16 | 0.03 | 13.55 | 0.04 |
| Nazari et al. [12] | 4.68 | 0.03 | 6.40 | 0.03 | 11.15 | 0.04 |
| Kool et al. [25] | - | - | 6.25 | - | 10.62 | - |
| RLCA-16 | 4.55 | 0.03 | 6.16 | 0.03 | 10.65 | 0.04 |
| OR-Tools [39] | 4.55 | 0.03 | 6.13 | 0.03 | 10.47 | 0.04 |
| Optimal | 4.55 | 0.03 | 6.13 | 0.03 | - | - |

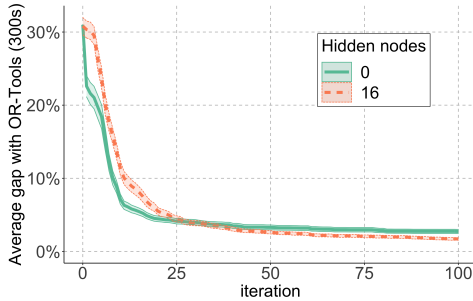

Figure 1: Results of our approach on 50 instances from the CVRP library (A and B) [27], ranging in size from 32 to 78 cities. We perform 100 policy iterations on each instance.

## 4.1 Comparison with existing methods and runtime analysis

In order to compare the results of our approach to existing RL algorithms, we consider random instances described in [12], in which cities are sampled uniformly at random from the unit square, distances are Euclidean, and demands are sampled uniformly at random from $\{1, \ldots, 9\}$. We present results on instances with 11, 21 and 51 cities in Table 1.

We note that our solutions compare favorably with both simple heuristics and existing RL approaches, and nearly match the performance of state-of-the-art operations research methods. We must, however, qualify the comparison with a caveat: existing RL approaches for CVRP [12, 25] consider a distributional setting where learning is performed over many instances and evaluated out of sample, whereas we consider a single-instance setting. Though these approaches are different, they nevertheless provide a useful comparison point for our method.

One advantage of our method is that it can be used on instances without distributional information, which may be closer to a real-world setting. In Figure 1, we evaluate our performance on standard library instances from CVRPLIB [27, 38], comparing our results with those of OR-Tools. Over 50 instances, the average final gap against OR-Tools is just 1.7%.

It is of interest to study our method's runtime. At training time, our two main operations are evaluating many sample paths with the current policy, and re-training the value function estimator (at evaluation time, we just evaluate one sample path). With up to 16 hidden ReLUs, neural net training is fast, so the bottleneck is solving (2). We cannot provide precise runtimes because our experiments were executed in parallel on a large cluster of machines (which may suffer from high variability), but we present average MIP runtimes given different architectures and solvers in Table 2 and, based on these values, we describe how to calculate an estimate of the runtimes. The runtime of a policy iteration is (# sample paths) × (# MIPs per path) × (MIP runtime). For $n = 21$ cities (16 hidden nodes), SCIP solves the average MIP in $\sim 3$s (Gurobi in $\sim 0.4$s), and we almost never exceed 10 MIPs per path, so

Table 2: Average time in seconds to solve a single action selection problem, with the MIP solvers Gurobi 8.1 and SCIP 6.0.2, averaged over 50 random instances. Instances are from the first policy iteration after taking a single random action, trained using 5000 path evaluations.

(a) $n = 21$

| Hidden nodes | Runtime (s) SCIP | Gurobi |
|---|---|---|
| 0 (LR) | 0.059 | 0.0085 |
| 4 | 0.84 | 0.16 |
| 16 | 3.1 | 0.38 |

(b) $n = 51$

| Hidden nodes | Runtime (s) SCIP | Gurobi |
|---|---|---|
| 0 (LR) | 3.19 | 0.097 |
| 4 | 132.5 | 11.9 |
| 16 | 234.7 | 39.3 |

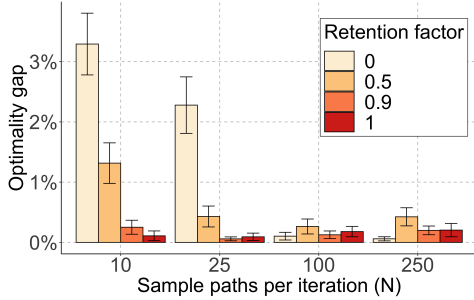

(a) Gap for varying amounts of data and retention factors after 30 policy iterations ($n = 11$).

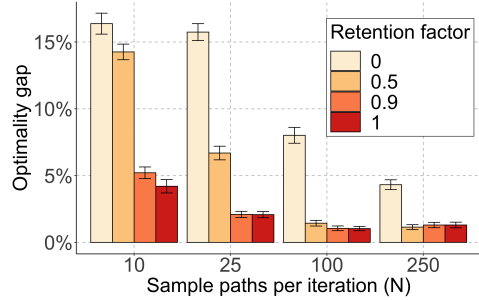

(b) Gap for varying amounts of data and retention factors after 30 policy iterations ($n = 21$).

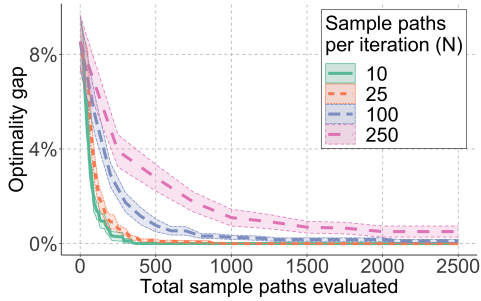

(c) Gap as a function of total evaluated sample paths, with retention factor $\gamma = 1$ ($n = 11$).

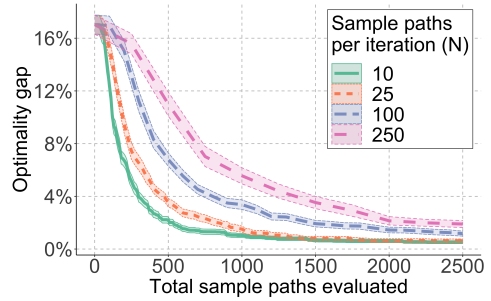

(d) Gap as a function of total evaluated sample paths, with retention factor $\gamma = 1$ ($n = 21$).

Figure 2: Effect of the amount of data on performance. Results averaged over 50 random Euclidean instances with 11 or 21 cities, error bars indicate standard errors (SEM).

computing a policy iteration with 250 sample paths takes about 2h using SCIP (15min using Gurobi). We can reduce runtime with parallelism: with as many machines as sample paths, the SCIP running time becomes about 30s (plus parallel pipeline overhead). For $n = 51$, SCIP is slower ($\sim 240$s per MIP), and a policy iteration may take up to an hour in parallel. In contrast, Nazari et al.'s [12] runtime bottleneck is neural net training (hours), but evaluation is much faster (seconds).

## 4.2 Ablation studies and sensitivity analysis

As a success metric, we evaluate the quality of a solution $x$ using the gap $g$ between $x$ and the provably optimal solution $x_{\text{CP-SAT}}$ obtained by CP-SAT on the same instance, where $g = (x - x_{\text{CP-SAT}})/x_{\text{CP-SAT}}$.

**Data requirements.** At each policy evaluation step, we evaluate the current policy from $N$ randomly selected start states to obtain sample paths. By evaluating more sample paths, we can use more data to update the policy, but we also require computing resources. In Figs. 2a and 2b, we show the solution quality obtained after 30 policy iterations, as we vary the number of sample paths evaluated per iteration $N$ and the retention factor $\gamma$. For a fixed number of iterations, results improve

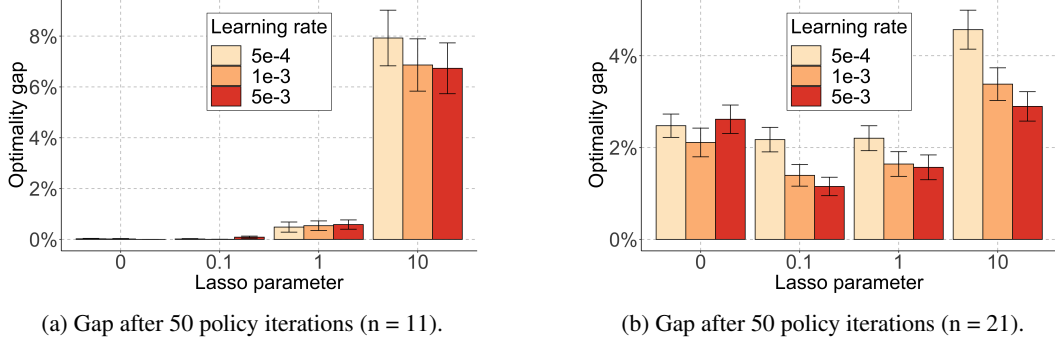

(a) Gap after 50 policy iterations (n = 11).

(b) Gap after 50 policy iterations (n = 21).

Figure 3: Effect of training parameters (lasso weight and learning rate) on performance. Results averaged over 50 random Euclidean instances (11 or 21 cities), error bars indicate standard errors (SEM).

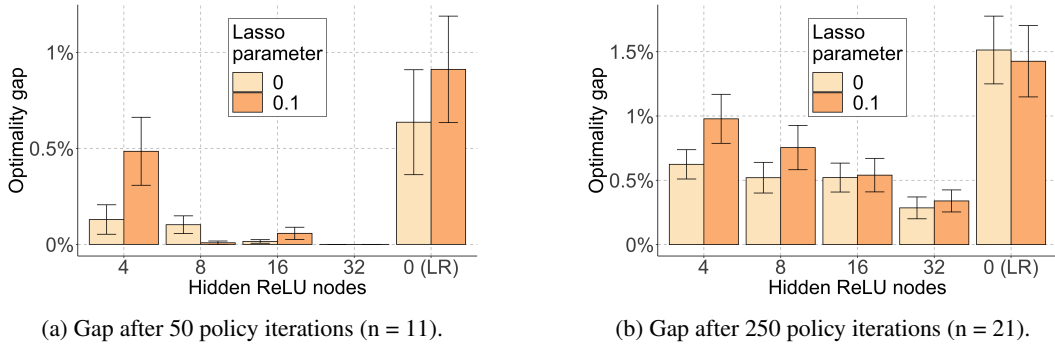

(a) Gap after 50 policy iterations (n = 11).

(b) Gap after 250 policy iterations (n = 21).

Figure 4: Effect of neural network size on performance (0 hidden nodes corresponds to linear regression). Results averaged over 50 random Euclidean instances, error bars show standard errors (SEM).

with more data, with diminishing returns. Keeping data between policy iterations significantly improves the solution quality, especially with few evaluations per iteration and little to no decay.

Even with just 25 evaluations per iteration, keeping data from one iteration to the next leads to solutions with a very small gap. In Figs. 2c and 2d, we show the solution quality when keeping data between iterations (with $\gamma = 1$) as a function of the cumulative number of evaluations and the number of evaluations per iteration. We obtain the best results with 250 iterations and 10 evaluations per iteration, suggesting that rapid iterations may be preferable on small instances.

**Architecture, training and regularization.** In Figure 3, we consider a neural network with 16 hidden ReLU nodes and show the effect of the batch SGD learning rate and the LASSO regularization parameter $\lambda$ (given all neural network weights as a vector $w$ of length $|w|$, we penalize the training objective with the LASSO term $(\lambda/|w|)\|w\|_1$). We notice that mild regularization does not adversely impact performance: this is particularly important in our setting, because a sparser neural network can make the action selection problem MIP formulation easier to solve. In Figure 4, we analyze the effects of increasing neural network size on the solver performance. We note that even 4 neurons leads to a significant improvement over a linear model; on small instances, a 32-neuron network leads to optimal performance.

**Combinatorial lower bounds.** As mentioned above, we augment the neural network modeling the value function with known combinatorial lower bounds. We can include them in the action selection problem, but also during training by replacing $\hat{V}(s)$ with $\max(\hat{V}(s), LB^1(s), \ldots, LB^P(s))$ in the training objective. Figure 5 shows that including these lower bounds provide a small but nonetheless significant improvement to the convergence of our policy iteration scheme.

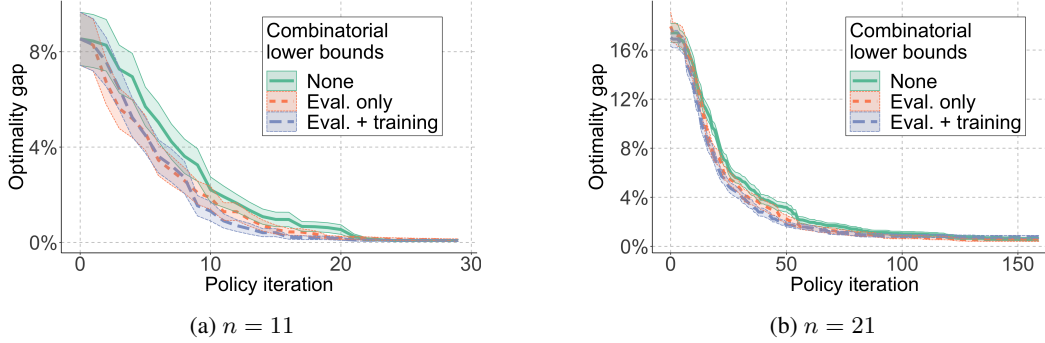

(a) $n = 11$                 (b) $n = 21$

Figure 5: Effect of including lower bounds in our RL framework. Results averaged over 50 instances.

## 5   Conclusion and Discussion

We have presented a novel RL framework for the Capacitated Vehicle Routing Problem (CVRP). Our approach is competitive with existing RL approaches, and even with problem-specific approaches on small instances. A combinatorial action space allows us to leverage the structure of the problem to develop a method that combines the best of reinforcement learning and operations research.

A big difference between this method and other RL approaches [12, 25] is that we consider a single instance at a time, instead of learning a single model for multiple instances drawn from a similar distribution. Single-instance learning is useful because the distribution of real-world problems is often unknown and can be hard to estimate because of small sample sizes. However, a natural extension of this work is to consider the multi-instance problem, in which learning is performed offline, and a new instance can be solved via a single sequence of PC-TSPs.

Another possible extension of this work is the consideration of other combinatorial optimization problems. A simple next step would be to impose an upper bound on the number of routes allowed, by augmenting the state space to keep track of the number of past routes and penalizing states where this number is too high. Beyond this simple example lies a rich landscape of increasingly complex vehicle routing problems [13]. The success of local search methods in tackling these problems suggests an orthogonal reinforcement learning approach, in which the action space is a set of cost-improving local moves, could be successful.

Beyond vehicle routing, we believe our approach can be applied to approximate dynamic programming problems with a combinatorial action space and a neural network approximating a cost-to-go or $Q$ function. Even the simpler case of continuous medium-dimensional action spaces is a challenge. In [22], a similar approach to ours was competitive with state-of-the-art methods on the DeepMind Control Suite [40].

## Broader Impact

This paper presents methodological work and does not have foreseeable direct societal implications.

## Acknowledgments and Disclosure of Funding

We would like to thank Vincent Furnon for his help in selecting OR-Tools parameters to significantly improve baseline performance.

## Footnotes

[1] In fact, the state transition model is deterministic and we have perfect knowledge of it.

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
