[Supplementary Material · neurips-final.pdf]

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

# A   Algorithms, hyperparameters, and code

In this section, we recap key hyperparameters in our approach, as well as their default values.

1. Overall parameters:
   - Number of policy iterations: we vary this in the paper, but our default value is 250.
   - Data retention factor: our default value is 1 (data from previous iterations is not decayed in the mean-squared error training objective).

2. Learning parameters:
   - Neural network architecture: we use a fully-connected neural network, with an input layer of size $n$ (binary input), a single hidden layer with 16 ReLU-activated nodes, and a single linear-activated output node.
   - Learning rate and batch size: we optimize the neural network parameters using standard batch stochastic gradient descent, with a fixed learning rate (default value $5 \times 10^{-4}$) and batch size (default value 10).
   - Epochs: we typically pass through the entire dataset 500 times.
   - LASSO regularization: given all neural network weights as a vector $w$, we penalize the training objective with the LASSO term $\lambda/ \cdot \|w\|_1 /$(number of weights), where the default value of $\lambda$ is 0.1. We note that when turning off LASSO regularization (not recommended), we can increase the learning rate to $1 \times 10^{-3}$ and decrease the number of epochs to 300, affording up to almost 2x training speedup. For models with fewer than 16 neurons, we do not use any regularization (i.e. $\lambda = 0$).

3. Evaluation parameters:
   - Number of sample paths to evaluate: we select $N$ random start states to initialize sample path computation. The larger the value of $N$, the more data we obtain to train the cost-to-go estimator. We typically set $N = 10$, a very small value since our ablation studies suggest there is more to gain from many policy iterations than from many data points.
   - Zeroing threshold: when solving the action selection problem, we integrate the neural network modeling the cost-to-go into a MIP formulation. MIP solvers can run into numerical issues in the presence of very small coefficients, so we round every coefficient with absolute value less than a threshold (typically 0.001) to zero. We have observed that this has no measurable effect on solution quality and greatly reduces the chance of the solver failing.
   - Time limit: we impose a time limit of 60 seconds on the MIP solver when solving the action selection problem. The solver will then return the best solution found so far (but not necessarily the optimal one). The time limit is rarely reached. On the large instances (the random instances with 51 cities and CVRPLIB instances), we use a time limit of 600 seconds.
   - Lower bounds: at the evaluation step, we can include simple combinatorial lower bounds. The default setting is to include them. We describe these lower bounds in more detail in a later section of this appendix.
   - Cuts: a key way to improve MIP runtime and solution quality is the addition of "cuts", inequalities that sharpen the solver's ability to prune the search tree using linear programming (LP) bounds. In the default configuration, we add cuts to sharpen the formulation of the neural network argmin problem following the approach detailed in [14]. We also follow a lazy constraint scheme for constraints (2f) and (2e) in which we add violated constraints at every node in the branch-and-bound tree rather than only at integer solutions. Finally, we use a linear programming preprocessing routine to update bounds.

The parameters above describe a setting in which each policy iteration is quick and we continuously improve the policy with a small amount of data. If parallel computing infrastructure is available, we consider a "high parallelism" mode, in which we increase the number of sample paths to 200 per iteration over 100 policy iterations.

The experiments on CVRPLIB and the 51 city random instances were run in high parallelism mode, while the ablations on 11 and 21 cities were run with the default settings (and the changes listed in each ablation).

**Algorithm overview.** Algorithm 1 presents an overview of our policy iteration scheme in algorithm block form. We note that this is a simplified description of our method, which does not include all refinements, e.g., the mechanism to conserve data from iteration to iteration.

---

**Algorithm 1** High-level overview of our policy iteration scheme. The inputs are simply the problem parameters, namely demands $d$, distances $\Delta$, capacity $Q$, and the number of policy iterations $K$.

---
1: **function** SOLVECVRP($d, \Delta, Q, K$)
2:      $\pi \leftarrow \pi^0$          ▷ Initialize policy
3:      **for** $k = 1$ to $K$ **do**
4:          $\mathcal{D} \leftarrow \emptyset$          ▷ Initialize empty dataset
5:          **for** $i = 1$ to $N_k$ **do**
6:              Select a random state $s_0$
7:              $\mathcal{D} \leftarrow \mathcal{D} \cup$ EVALUATEPOLICYFROMSTATE($\pi, s_0$, `false`)      ▷ Add new data
8:          Use dataset $\mathcal{D}$ to train a new value function approximation and update the policy $\pi$
9:      Define $s_0$ as the state where all cities are unvisited
10:      **return** EVALUATEPOLICYFROMSTATE($\pi, s_0$, `true`)
11: **function** EVALUATEPOLICYFROMSTATE($\pi, s_0, x$)
12:      $c \leftarrow 0, s \leftarrow s_0$
13:      **while** $s$ is not terminal **do**
14:          $a \leftarrow \pi(s)$
15:          $c \leftarrow c + C(a)$
16:          $s \leftarrow T(s, a)$
17:      **if** $x =$ `true` **then**
18:          **return** the list of selected actions (routes) and the total incurred cost
19:      **else**
20:          **return** all visited states and the total cost incurred from each one, denoted $\{s^i, c^i\}_i$

---

Our code is available as part of the `tf.opt` repository: https://github.com/google-research/tf-opt.

**Optimizing over a trained neural network.** The second term of the objective (2a) is a nonlinear function (fully-connected one-layer neural network with ReLU activations), and nontrivial to model using mixed-integer linear programming. We use a technique developed by Anderson et al. [14] to model $\hat{V}(t)$. For clarity, assume that $\hat{V}(\cdot)$ has a single hidden layer, with $P$ hidden nodes, each with a ReLU activation.

Let $w^{(p)} \in \mathbb{R}^n$ designate the vector of weights, and $b^{(p)} \in \mathbb{R}$ the bias term, for the $p$-th hidden node. Define $w^{\text{output}} \in \mathbb{R}^P$ and $b^{\text{output}} \in \mathbb{R}$ analogously for the output layer. For any vector of weights $w$, let supp($w$) indicate the set of indices $i$ such that $w_i \neq 0$. Finally, define $\rho : \mathbb{R} \to \{0, 1\}$ a modified version of the sign function, where $\rho(a)$ returns 1 if $a \geq 0$, and 0 otherwise. Then we can write the

problem $\min_{t\in\{0,1\}^n} \hat{V}(t)$ as

$$\min \quad \hat{V}(t) \quad := \sum_{p=0}^{P-1} w_p^{\text{output}} y^{(p)} + b^{\text{output}} \tag{3a}$$

$$\text{s.t.} \quad y^{(p)} \quad \geq w^{(p)} \cdot t + b^{(p)} \qquad \forall 0 \leq p < P \tag{3b}$$

$$y^{(p)} \quad \leq w^{(p)} \cdot t + b^{(p)} - M_-^{(p)}(1 - z^{(p)}) \qquad \forall 0 \leq p < P \tag{3c}$$

$$y^{(p)} \quad \leq M_+^{(p)} z^{(p)} \qquad \forall 0 \leq p < P \tag{3d}$$

$$y^{(p)} \quad \leq \sum_{i \in I} w_i^{(p)} \left( t_i - \left( 1 - \rho\left( w_i^{(p)} \right) \right) \left( 1 - z^{(p)} \right) \right) + \tag{3e}$$

$$\left( b^{(p)} + \sum_{i \notin I} w_i^{(p)} \rho\left( w_i^{(p)} \right) \right) z^{(p)} \qquad \forall 0 \leq p < P, I \subseteq \text{supp}\, w^{(p)} \tag{3f}$$

$$t \quad \in \mathcal{T} \tag{3g}$$

$$y^{(p)} \quad \in \mathbb{R} \qquad \forall 0 \leq p < P \tag{3h}$$

$$z^{(p)} \quad \in \{0, 1\} \qquad \forall 0 \leq p < P. \tag{3i}$$

Notice that in addition to the $n$ initial binary decision variables $t_i$, we define two additional decision variables for each hidden node in the neural network. One is continuous and models the output of the hidden node, the other is binary and indicates whether the pre-activation function is positive or negative (i.e. whether the ReLU is active or not). This relationship is enforced by the "big-$M$" constraints (3c) and (3d). In principle, any large enough values of $M_+$ and $M_-$ can enforce this relationship, but values that are too large weaken the formulation and can decrease tractability. We therefore compute these big-$M$ values for each hidden node by maximizing (or minimizing, depending on whether computing $M_-$ or $M_+$) the pre-activation function over the LP relaxation of the neuron's input domain (which includes all the problem specifications, including the PC-TSP and knapsack constraints). This operation can be performed iteratively, neuron-by-neuron, to compute suitable values for $M_+^{(p)}$ and $M_-^{(p)}$.

The set $\mathcal{T}$ encapsulates any other constraints on the binary variables $t$ (input domain of the neural network $\hat{V}(\cdot)$), in particular the ones associated with the prize-collecting traveling salesman formulation (2).

We note that this formulation is not polynomial in size, as there are exponentially many constraints of type (3f). However, these constraints are not needed for correctness, they simply strengthen the formulation. When solving the action selection problem, we generate these constraints on the fly, using a linear-time separation oracle as described in [14].

**Local search warm start.** In order to speed up the MIP solve in the action selection problem (2), we provide the solver with an initial primal-feasible solution, obtained via a simple local search heuristic. This warm start ensures we always obtain a feasible solution and allows the solver to spend more time in the branch-and-bound tree. As a result, we can set a lower solver time limit, and increase the number of policy iterations performed in a fixed amount of time.

Our local search heuristic (1-OPT with random start) can be described as follows. We first create a random feasible route, by randomly sampling unvisited cities until we hit the capacity constraint. We then consider every unvisited city, and compute the cost of removing this city in the route (if it is already included) or including it in the route (if it is not). For each of the $\mathcal{O}(n)$ unvisited cities, evaluating the cost requires one call to a TSP solver (to evaluate the route distance) and one neural network evaluation (to evaluate the future cost of unvisited cities). Both are computationally tractable, and allow us to quickly perform many iterations, possibly with several random restarts.

**Combinatorial lower bounds.** In the main text, we mention refining the cost-to-go $\hat{V}(\cdot)$ with combinatorial lower bounds linear in the decision variables of problem (2). We now describe each bound in more detail – note that they each assume the distances respect the triangle inequality (metric vehicle routing):

1. Maximum out-and-back bound: given a state $s$, the cost-to-go cannot be exceeded by the distance to and from the furthest city from the depot, i.e.,
$$V(s) \geq \max_{i:s_i=1} (\Delta_{0i} + \Delta_{i0}).$$
   This bound is rather weak for a large number of remaining cities, but it is tight when a single city remains.

2. Shortest-edges bound: given a state $s$, we must take at least one edge into and out of each city, thus paying at least half the cost of the shortest edges into and out of each city, i.e.,
$$V(s) \geq \sum_{i:s_i=1} \min_{j:s_j=1;j\neq i} \frac{1}{2}(\Delta_{ji} + \Delta_{ij}).$$

3. Refined shortest-edges bound: we can refine the bound above using demand information to incorporate a bound on the minimum number of vehicles required (and thus the minimum number of times the depot must be visited), i.e.,
$$V(s) \geq \sum_{i:s_i=1,i>0} \min_{j:s_j=1;j\neq i} \frac{1}{2}(\Delta_{ji} + \Delta_{ij}) + \frac{1}{2}\left\lceil \frac{\sum_{i:s_i=1,i>0} d_i}{Q} \right\rceil \min_{j:s_j=1;j>0} \Delta_{0j}.$$

## B CVRP instances

In this section, we describe the CVRP instances we use to evaluate our reinforcement learning framework.

**Random Euclidean instances.** We follow the generation procedure of Nazari et al. (2018) to construct random instances. We sample $n$ locations uniformly at random in the unit square, then define the distance $\Delta_{ij}$ to be the Euclidean distance from city $i$ to city $j$ (symmetric). One of the cities is randomly selected to be the depot, and the demand for the remaining $n-1$ cities is uniformly sampled from $\{1, 2, \ldots, 9\}$. The vehicle capacity $Q$ scales with the number of cities, with $Q = 20$ for $n = 11$, $Q = 30$ for $n = 21$, and $Q = 40$ for $n = 51$.

**Standard library instances.** An issue with evaluating algorithms uniform Euclidean vehicle routing instances is that such instances are known to satisfy certain strong regularity properties that may not be manifested in real-world problems [41, 42]. In an effort to benchmark methods against more realistic instances, the CVRP library (CVRPLIB) is an online resource cataloging instances from the literature—either real problems, or synthetic examples inspired by certain properties of real-world instances. We include 50 instances from CVRPLIB ("A" and "B" instances, ranging in size from 32 to 78 cities) [38]. We note that in many cases, the optimal values for these problems are known, and listed on the CVRPLIB website [27]. However, we do not use these optimal values because they consider a slightly different setting in which the number of vehicles is fixed, and thus overestimate the true optimum in our unbounded-fleet setting.

## C Baselines

In this section, we briefly describe the baselines against which we compare our results.

**OR-Tools.** Google's open-source OR-Tools library is an often-used benchmark for combinatorial optimization problems, and incorporates one of the best existing vehicle routing solvers [39], which combines exact approaches with local search and other heuristics to provide high-quality solutions. On instances of moderate size such as the ones presented in this paper, OR-Tools configured to perform a few minutes of local search typically produces optimal or near-optimal solutions, motivating our use of OR-Tools as a reference point when an optimal approach does not scale. The results we

report with OR-Tools are considerably better than those reported in previous reinforcement learning for vehicle routing papers [12, 25]. This reflects the fact that we configure OR-Tools for solution quality and not for speed, in contrast to other methods.

**Optimal.** Vehicle routing problems can be solved to optimality up to a certain problem size using mixed-integer programming (MIP) or constraint programming (CP) approaches. In addition to a routing solver, the OR-Tools library also provides a constraint programming solver called CP-SAT. To avoid confusion and remain consistent with the literature on reinforcement learning for vehicle routing, we refer to the routing solver as "OR-Tools" and the CP-SAT solver as "CP-SAT", even though both are technically part of OR-Tools.

We use CP-SAT to compute both feasible solutions and lower bounds. On small instances, the lower bounds coincide with the best feasible solution, yielding a certificate of optimality. When tractable, these provably optimal solutions are a reference point for our results, and we use the gap to the optimal solution as a key metric of success. Unfortunately, CP-SAT cannot prove optimality in a reasonable time for $n = 51$ cities. For larger instances, we therefore fall back to the near-optimal OR-Tools (routing) solver as a reference point. We note that as CP-SAT and the OR-Tools routing solvers only work on integer distances, we multiply all problem distances by $10^4$ and round to the nearest integer, so results are accurate within $10^{-4}$.

**Greedy.** One of the claims of this paper is that our method performs well on CVRP instance both because of the combinatorial structure of the action space and because of the machine learning model's ability to learn the complexity of the value function. A useful benchmark for our approach is therefore a method which features only the combinatorial action space without the learning component. This approach involves solving problem (2), replacing the cost-to-go $\hat{V}(t)$ with the following trivial upper bound:

$$UB(t) = \sum_{i=0}^{n-1}(\Delta_{0i} + \Delta_{i0})t_i,$$

representing the total distance of covering all remaining cities with one route per city. We refer to this baseline as a greedy method, because it overestimates the cost-to-go and thus compels the optimal action to pack as many cities as possible (and particularly cities far from the depot).

**Existing approaches.** Finally, we compare our approach with existing RL approaches for vehicle routing [12, 25]. This comparison is less direct than the two above, because both approaches consider a multi-instance learning setting, where the model is trained on a large sample of problem instances and evaluated on unseen instances from the same distribution. As a result, both papers [12, 25] report out-of-sample solution quality metrics, whereas we train and evaluate on one instance at a time. Both frameworks are valuable: on one hand, learning insights from multiple instances that can be generalized to a new problem can significantly improve solve times; on the other hand, real-world vehicle routing problems do not typically come with distributional information, rendering prior learning irrelevant. Though they are not directly comparable, we nevertheless display solution quality metrics for both of these methods along with ours, as evidence that our framework produces solutions in the same ballpark as other RL frameworks.

# D Additional simulation results

In Section A of the appendix, we included a description of our hyperparameters, and in particular identified a "high parallelism" setting. We now provide some experimental justification for this setting in Figure 6. We see that increasing the number of sample paths per iteration can provide significant improvements on a per-iteration basis. Computing sample paths is a highly parallelizable task, so depending on the computing platform available, it may be more efficient to consider a small number of policy iterations, and compensate by computing many sample paths in parallel at each iteration.

(a) $n = 11$

(b) $n = 21$

Figure 6: Effect of number of sample paths per iteration on the gap at each iteration ($\gamma = 1$). Results averaged over 50 random Euclidean instances, error bars show standard errors (SEM).