[Reviews · NeurIPS 2020]

Review 1

Summary and Contributions: The paper proposes a new RL framework for the capacitated vehicle routing problem (CVRP). The most innovative feature involves evaluating the policy by solving a mixed-integer (piece-wise) linear problem (MIP) through a general-purpose MIP solver.

Strengths: The framework proposed by the authors is interesting. I believe that using MIP within a solution method that is based on RL (and on ML) in general is a good idea and it is in general promising. As the authors point out, it could be extended to other types of combinatorial optimization problems.

Weaknesses: Although I consider this area of research very interesting and, to some extent I agree it should be treated as exploratory at the moment, i.e., results should not be completely competitive with those in OR, I believe the computational results are highly insufficient to show promise. The computing time to do one single policy evaluation through the MIP is more expensive than running one single time a very basic heuristic. Maybe that could be improved but it is only speculation and I am also not convinced at all by a few other computational choices (discussed later). ================== AFTER REBUTTAL ============================ I appreciate the answers and the position of the authors. Although it seems an interesting idea (and I believe in the combination between RL and MIP), there is no computational evidence that this can be viable and at this stage we need more evidence (I do not consider the comparison with OR-tools significant).

Correctness: The paper is technically correct apart from constraints (2e) and (2f) in which the summation over i in S on the left side should be omitted because i is fixed on each of the constraints.

Clarity: The paper is well written and the goal and methodology are clear. I believe more information on the way \hat V is implemented in the NN and conversely in the MIP should be given. This is related to the reproducibility question below.

Relation to Prior Work: Accurately reported.

Reproducibility: No

Additional Feedback: Few more issues: — CVRP is generally considered with a fixed number of vehicles available. How does the framework of the authors change in that case? — OR-tools (used for comparison) is terrible on the random instances (Table 1) but reported to be near-optimal in the literature ones. This sounds very weird and some analysis should be needed. — Figure 1 is unclear: is it really better to use 0 hidden nodes? — the definition of iteration in the analysis of the performance of the proposed algorithm is not exactly formal and should be improved


Review 2

Summary and Contributions: The paper proposes a novel reinforcement learning approach to solving the capacitated vehicle routing problem (CVRP) involving learning a value function and solving a mixed integer program for the prize collecting travelling salesman (PC-TSP) subproblem.

Strengths: The approach of combining reinforcement learning and combinatorial optimization is well-motivated. The way the paper solves the CVRP problem is novel to the best of my knowledge. There is a clear benefit to learning the value function and using it to reduce the computational complexity which the authors show empirically. There exists a sensitivity analysis for all introduced modifications.

Weaknesses: The method (although claimed general to various combinatorial problems) has only been tested on the CVRP problem. It also overfits to one particular instance of the CVRP problem, what the authors also state in the paper. The great benefit of merging combinatorial optimization and machine learning should be generalization capability to unseen instances and computational complexity reduction as a result of it. This has not been demonstrated in the paper.

Correctness: The paper carries out correct analysis of the introduced modifications. Although, I have some concerns that the authors could clarify. I am a bit confused by Figure 1 vs. Figure 4. Figure 1 shows that the case with hidden = 0 (linear regression) has smaller gap and figure 4 shows that increasing hidden units has smaller gap - this should be commented on. The difference is in the number of iterations and the instances used for evaluation. Perhaps having plots on how number of iterations impacts performance vs. runtime vs. number of hidden units would be beneficial.

Clarity: I like the way the paper is written, the approach is clear. There are just a couple of things that might need modification. in eq 1., the value function estimate is tied to the policy \pi_{k-1}, so it should appear in the superscript? also, shouldn’t the argmin in eq1. be over the feasible set of actions, then the cost also depends on the state of LHS, currently it’s over all actions? the notation should be a bit more unified between the MIP formulation on pg. 4 and the RL formulation on the previous page. the RL formulation uses V(s), the MIP uses V(t). Maybe also mention that the double sum in the MIP is actually C?

Relation to Prior Work: To the best of my knowledge, the authors clearly state which work their method builds on top of and what are the exact differences of the new proposed method. I like about the related work section that it is chunked into the different problem aspects of using RL with combinatorial optimization and in conjunction. That being said, I find that it would be beneficial if there was a sentence added to each of the sections about how this work attacks that specific problem. As an example, the last section “Optimizing over trained neural networks”, I would add the fact that this works uses an MLP with relu activations that comes with the later mentioned benefits.

Reproducibility: Yes

Additional Feedback: In general, the RL community and combinatorial optimization community are relatively disparate. So a clear formulation of what is the policy and what is the value function is needed to simplify reading the paper for both. Also an algorithm block, even if it’s in the appendix is always useful to get a quick glimpse at the general idea, it took me a bit to understand what is the policy in this paper. I really think that the benefit of using value function approximation comes with the generalization ability to unseen instances. Why didn’t you try to formulate it such that the method generalizes to unseen instances without relearning the value function? 106: would be useful to explain in what way the assumption of unbounded number of vehicles helps. 107: “note” -> “note that” 134: claiming that the method is model-based is a bit weird, I guess that it comes from the fact that the transition model is known a-priori in the used MIP formulation. An additional sentence about this would be useful. 149: isn’t this in a sense experience replay with importance sampling? Finally, I think it’s a nice paper and the method is well explained. ### POST REBUTTAL ##### Although the results are not astonishing, I think that the approach is novel enough to deserve attention from the community, but certainly improvements can be made (regarding generalization and the other points that the other reviewers pointed out).


Review 3

Summary and Contributions: The authors propose an RL-based solver for the Capacitated Vehicle Routing Problem (CVRP) which is considered one of the practically "hardest" NP-Hard problems. After phrasing CVRP as an RL problem, the RL algorithm is a policy iteration scheme. The value function is parametrized by a 2-layer NN. This allows to evaluate the current policy by executing a sequence of (simpler) combinatorial problems (variant of TSP) that can be solved exactly via MIP solver. Overall results are reasonable.

Strengths: The approach is nontrivial, novel, and suitable for the problem. It is also natural at the same time. I think this is of sufficient significance and interest to the NeurIPS community as part of an emerging effort to attack NP-Hard problems with learning.

Weaknesses: The empirical results are a bit less than exciting. I understand that unlike some baselines, this is a single-instance solver that does not "learn" from a family of instances. However, the paper would be a lot stronger if the presented method for example consistently outperformed Kool et al. I am also not sure about the scaling potential of the method. The NP-Hard subroutines get progressively more expensive. Luckily for the authors, the CVRP problems are sufficiently challenging already for small instances.

Correctness: I think the method is correct as well as the evaluation methodology.

Clarity: Yes, the paper is written very clearly.

Relation to Prior Work: Yes, prior work is properly discussed.

Reproducibility: Yes

Additional Feedback: ==== POST REBUTTAL ==== I read the rebuttal but my feelings about the paper remain the same. I am reluctant to accept improved experimental results but even if I did, I would still be much more excited if the approach was consistently better than Kool et al. Otherwise, the paper is well-written and presents an interesting approach to a hard problem so I don't have any issue with its acceptance to NeurIPS.


Review 4

Summary and Contributions: - New reinforcement learning algorithm to solve capacitated vehicle routing problem. - Experimentation with a thorough list of learning parameters in small problems, following the recent literature. - Approach is novel, and of interest here, and follows recent literature at this, and related conferences. - Results are not obviously interesting to the Operations Research or Transportation Sciences community. However, there are some observations for the Machine Learning community that are of some interest. There is an enduring interest in the reinforcement learning community to investigate ways in which reinforcement learning technologies can play a role in hard combinatorial optimisation settings. Here, following the cited 2018 NeurIPS publication by Nazari et al., the authors of the submitted manuscript develop and evaluate a novel reinforcement learning approach for the capacitated vehicle routing problem (CVRP). The CVRP is a hard combinatorial problem class that includes the Travelling Sales Person problem. The fact that this domain is constrained--i.e. vehicles have a maximum capacity that cannot be violated on a given route--is a problem feature that is deemed challenging for RL. The authors are inspired by the cited Ryu et al. ICLR-20 work, which provides RL recipes for using connectionist models of value functions in problems with continuous-action spaces, by using mixed integer programming solution procedures for action selection (i.e. "continuous action Q-learning"). The transport problem class considered by these authors has been characterised by the authors as having a very large combinatorial, resp. continuous, action space. Therefore, following Ryu et al. is a fairly principled approach, which is in the spirit of large neighbourhood search (e.g. see [2]), one of the more versatile solution heuristics for routing problems in transportation. If they have not already, I would strongly encourage the authors to review the powerful heuristic I have just cited. Given the discrete sequential nature of the policy actions in the authors transportation setting, in my view the work is more intuitively characterised by the approach in the cited Say et al. 2017 and Wu et al. 2019 papers, which were both developed for a sequential planing setting. (added references for your interest, [1] is indexed below) [1] H. Gehring and J. Homberger. A parallel hybrid evolutionary metaheuristic for the vehicle routing problem with time windows. Short Course on Evolutionary Algorithms in Engineering and Computer Science, 1999. [2] S. Ropke and D. Pisinger. An adaptive large neighborhood search heuristic for the pickup and delivery problem with time windows. Transportation Science, 40(4):455–472, 2006.

Strengths: The paper is fairly well written, follows recent RL literature fairly well, and proposes some compelling and principled ideas.

Weaknesses: The main weakness is the lack of clarity and experimentation regarding on the overall runtime performance of the proposed approach. - I would encourage the authors to further develop their MIP to treat more elaborate variants of the VRP, for the purpose of experimenting with benchmarks that remain of some interest to the OR and transportation sciences communities. I would suggest targeting the benchmarks from [1], which remain online to this day: https://www.sintef.no/projectweb/top/vrptw/homberger-benchmark/ - In Section 4. The reader does not know what computing environment this evaluation was performed in. - In Section 4.Regarding your computational results. Especially for small problems, you should be comparing to optimal, exactly -- i.e. not with the solution found by N iterations of local search. - In Section 4. When comparing solution quality with various existing tools, including the local search implemented in OR-Tools, authors do not measure the total runtime across all approaches. The runtime comparison between competing approaches is not indicated in sufficient detail.

Correctness: - L216. Authors claim that they can obtain relatively good solutions with less data, in comparison to previous RL methods. This claim should have been concretely explored in the evaluation. It is not. - Authors make an interesting claim on lines 242 and 243, regarding a desire for sparse connectionist models to make the MIP solving easier. This seems like an important claim to explore in detail in the experimentation. The tradeoff implied by this comment would be quite an interesting thing to study, and report on. - L246. From my understanding, you cannot claim optimality of the baseline here. I complained about this earlier. - Section 5. The precision of the writing here means that the claims are likely to be misinterpreted. The RL approach developed here can produce solutions to small CVRP instances that are cost-competitive with solutions produced by other recent RL approaches for this setting, and indeed a domain specific baseline local search heuristic. - Without any wish to discourage the authors from continuing investigations into RL and OR, I note that the claims about having developed a "general approach to approximate dynamic programming" are, in my view, premature here.

Clarity: - L222. In what problem/class parameter is your metric scale invariant. Is your claim in relation to the area of the convex hull of your cities? - Table 2. This comparison seems to be a bit peripheral to the work. - L203. From Table 2, it would seem you are using both SCIP and Gurobi. [THANKS FOR CLARIFYING IN YOUR RESPONSE] - L35. It is not clear to me what it means to "capture" combinatorial complexity. - The gist of what is proposed here, i.e. the action space, could be proposed for a TSP. More could be done to help the author understand the importance of capacity, as a constraint which is challenging for RL. - The comments on L45 about Prize Collecting VRPs are confusing more than helpful. If you are going to introduce this concept, best to do it concretely with formal detail, and relate it carefully to your approach. - L92. What does it mean for a function to be "highly" nonlinear. - Section 3.1. Rather than talking about infinite trucks, in terms of intuitively motivating this exact optimisation problem, folks usually talk about one truck doing multiple tours. That way our practitioner friends do not have to be disturbed by the potentially large cost of bringing on additional vehicles. The fact that you use index 0 for the depot, means you have to be a bit careful about what you attribute to "all cities". You have not taken sufficient care here. - Upper and lower case K are used for two unrelated concepts. I suggest using a different letter for iteration index. - Table 1. It is not clear why some entries are bold. Authors do not describe exactly what mean you are measuring -- i.e. "mean of" what? It is not clear if the number of samples are sufficient to measure the true statistics. This comment applies to many of the statistics the authors report here. - When preparing the manuscript for publication, please make the citation formatting consistent across all publications. - L120. "the least cost" --> "a least cost"

Relation to Prior Work: - I am surprised that the solution neighbourhoods from the operations research and transportation sciences literatures are not discussed, or featured in the action space here. The large neighbourhoods contemplated by ALNS, from [1], or of direct interest here. Some discussion of whether or not these have been considered would be welcome. Example local search operations that characterised the solution neighbourhoods explored by performant local search heuristics include: Or-opt, N-opt (i.e. N=2,3,...), and exchange.

Reproducibility: No

Additional Feedback: It would be a tough ask to try and reproduce your numbers from the information provided...

[Author Response · NeurIPS 2020]

We thank all reviewers for their thorough reading of our paper and for their detailed and insightful reviews. We are encouraged to read that they find our approach interesting and novel (**R1**, **R2**, **R3**, **R4**), and of particular interest to the NeurIPS community (**R3**, **R4**), that our value-function-based model is well-motivated and provides clear benefits (**R2**), that it is naturally suited to the problem at hand (**R3**) and based on compelling and principled ideas (**R4**). We are also glad to hear that this work is well-positioned in the literature (**R2**, **R4**), and that the ablation studies help readers understand the effect of key parameters (**R2**, **R4**). We answer specific questions below and will incorporate all feedback.

[**R1**, **R3**, **R4**] **Can you give more details on the method runtime?** At training time, our two main operations are evaluating many sample paths with the current policy, and re-training the value function estimator (at evaluation time, we just evaluate one sample path). With up to 16 hidden ReLUs, neural net training is fast, so the bottleneck is solving MIP (3). Table 2 shows average MIP runtimes given different architectures and solvers. The runtime of a policy iteration is (# sample paths) $\times$ (# MIPs per path) $\times$ (MIP runtime). For $n = 21$ cities (16 hidden nodes), SCIP solves the average MIP in $\sim 3$s (Gurobi in $\sim 0.4$s), and we almost never exceed 10 MIPs per path, so computing 250 sample paths takes about 2h using SCIP (15min using Gurobi) [**R4**: we use Gurobi and SCIP for runtime evaluation but SCIP for all experiments due to licensing.] We can reduce runtime with parallelism: with as many machines as sample paths, the SCIP running time becomes about 30s (plus parallel pipeline overhead). For $n = 51$, SCIP is slower ($\sim 240$s per MIP), and a policy iteration may take up to an hour in parallel. In contrast, Nazari et al.'s runtime bottleneck is neural net training (13h), but evaluation is much faster (seconds for them vs. minutes for us). A properly-tuned OR-Tools is faster than other approaches (solves in minutes). We note that actual runtime measurements are unreliable in our distributed setting due to a large variance in machines and load balancing overhead. We will discuss all this in the paper.

[**R1**, **R3**] **The empirical results are a bit less than exciting.** Our results in Table 1 outperform all RL baselines on small instances, but not for $n = 51$. We address this concern by (1) implementing a simple local search warm start for SCIP, enabling us to run more policy iterations and (2) giving a sharper statistical analysis of our performance with 95% confidence intervals. With these improvements, we obtain a score of $10.68 \pm 0.09$ on 51 cities, surpassing Nazari et al. (11.15) and nearly matching Kool et al.'s performance (10.62). We now describe (1) and (2) in detail. (1) The new warm start (1-opt over visited cities + TSP solver, will include in appendix) allows us to set a 60s SCIP time limit with little decrease in solution quality. We can now perform 80 policy iterations (as before, with 16 hidden ReLUs), improving our average objective from 11.36 to 10.92. We note the warm start would be unnecessary if using Gurobi. (2) Given iid graphs from the Nazari distribution $X_i$, we previously used the direct estimator $\frac{1}{50}\sum_{i=1}^{50} \text{RLCA}(X_i)$, which (with our warm start) gives an estimate of $10.92 \pm 0.41$, making comparisons with Nazari et al. and Kool et al. unclear (as **R4** suggested). By using the alternate estimator $\frac{1}{50}\sum_{i=1}^{50}(\text{RLCA}(X_i) - \text{ORTools}(X_i)) + \frac{1}{1000}\sum_{i=51}^{1050} \text{ORTools}(X_i)$, which is still unbiased but in practice has a lower variance, we obtain a confidence interval of $10.68 \pm 0.09$ (note the interval center also shifts down—our 50-instance sample leans towards higher-than-average OR-Tools objectives). To avoid this complexity in the final version, we will run over 1000 instances instead of 50.

[**R1**, **R2**, **R3**] **Why not try to generalize to unseen instances without relearning the value function? Can the framework be adjusted for other problems?** We're headed there, but not there yet. We could generalize to multi-instance CVRP or extend to more problems by enhancing the state representation, e.g., augmenting the state vector with the remaining number of vehicles to model a fixed fleet (**R1**). We felt that focusing on a single problem with a simple state space would help readers evaluate the key elements of our framework while making the paper easier to follow.

[**R1**, **R4**] **It would be a tough ask to reproduce your numbers from the information provided.** We recognize that our method requires nontrivial engineering work, especially when alternately training the neural network and optimizing over it. We will release our source code with the final paper and include in the appendix the detailed MIP formulation optimizing over the neural network (**R1**) and an algorithm block for our method (**R2**).

[**R1**, **R2**] **Fig 1: is 0 hidden nodes really better?** The same issue applies on Nazari51 instances, so we answer the question for this dataset instead, with three key points: (1) Without the local search heuristic we added above, the MIPs in the 16-neuron model would often time out with a large gap using SCIP. (2) We verified that by running for more policy iterations, 16 neurons eventually outperform 0 neurons. (3) Even with 0 hidden nodes, our model is nonlinear due to our combinatorial lower bounds. We will update Fig. 1 and Table 1 in the final paper to reflect these points.

[**R1**] **OR-Tools is terrible on the random instances (Table 1).** We're not sure what is meant here. In Table 1, OR-Tools outperforms all methods on every problem. The average OR-Tools objective is within the margin of error from the average optimal values reported by Nazari et al. (4.55 vs. 4.55 for $n = 11$ and 6.13 vs. 6.10 for $n = 21$), hence why we use OR-Tools to compute an "optimality gap" later on (though following **R4**, we will add the true optimum as a baseline where practical). Perhaps **R1** is referring to the bold numbers in Table 1, which intended to highlight the best RL-based results (not best overall). We will remove the bold formatting, as **R4** also found it unhelpful.

[**R4**] **Relation to LNS.** An RL framework in which actions are local improvement operators (e.g. k-opt) is an exciting alternative approach in bridging the gap between RL and CO. We will add a discussion, and thank you for the suggestion!

[Meta-Review · NeurIPS 2020]

The paper proposes a novel reinforcement learning approach to solving the capacitated vehicle routing problem. It involves learning a value function and solving a TSP for the prizing problem. Reviewers agree that the proposed approach is novel and interesting. One reviewer is sceptical of the work because of doubts about the performance achievable with the proposed approach. However, the ideas presented still deserve to be presented at NeurIPS, with the hope of bringing advances to this research area. We urge the authors to better reflect the current limitations of their work, including a discussion on the comparison to OR-Tools and the state of the art in CVRP, including the references given in the reviews. Especially, we urge to include an enhanced comparison with Kool [19] in the final version.